# The Impact of Metal and Heavy Metal Concentrations on Vancomycin Resistance in *Staphylococcus aureus* within Milk Produced by Cattle Farms and the Health Risk Assessment in Kurdistan Province, Iran

**DOI:** 10.3390/ani14010148

**Published:** 2024-01-02

**Authors:** Yeganeh Sadeghian, Mahdieh Raeeszadeh, Hiva Karimi Darehabi

**Affiliations:** 1Graduate of Faculty of Veterinary Sciences, Sanandaj Branch, Islamic Azad University, Sanandaj 618, Iran; ysadeghian76@gmail.com; 2Department of Basic Sciences, Sanandaj Branch, Islamic Azad University, Sanandaj 618, Iran; 3Department of Food Hygiene and Public Health, Sanandaj Branch, Islamic Azad University, Sanandaj 618, Iran; hiva60iran@yahoo.com

**Keywords:** vancomycin, toxic heavy metals, *Staphylococcus aureus*, antibiotic resistance, raw milk

## Abstract

**Simple Summary:**

This study focuses on ensuring food safety and hygiene, investigating vancomycin resistance in raw milk from livestock farms in Kurdistan. The research, involving 160 milk samples, explores the correlation between resistance and heavy metals. Our findings reveal that 8.75% of samples contain *Staphylococcus aureus*, with 28.58% exhibiting vancomycin resistance. Significant variations in arsenic, iron, zinc, sodium, and aluminum concentrations were noted between resistant and sensitive samples. Elevated arsenic, iron, and aluminum, coupled with decreased zinc, showed a significant association with vancomycin resistance. The lead, cadmium, mercury, zinc, and iron levels exceeded permissible limits. The Target Hazard Quotient (THQ) for cadmium indicated a high non-carcinogenic risk, while the Target Risk (TR) for arsenic fell within the carcinogenic range. Accumulation of heavy metals could impact antibiotic resistance in milk, stressing the need to control arsenic residues for national safety.

**Abstract:**

In today’s food landscape, the paramount focus is on ensuring food safety and hygiene. Recognizing the pivotal role of the environment and its management in safeguarding animal products, this study explores vancomycin resistance in raw milk from livestock farms in the Kurdistan province and its correlation with metal and heavy metal. One hundred and sixty raw milk samples were collected from various locations, with heavy metal concentrations analyzed using ICP-MS. Identification of *Staphylococcus aureus* and vancomycin resistance testing were conducted through culture and the Kirby–Bauer method. This study investigates the relationship between resistance and heavy metal levels, revealing that 8.75% of milk samples contained *Staphylococcus aureus*, with 28.58% exhibiting vancomycin resistance. Significant variations in arsenic, iron, zinc, sodium, and aluminum concentrations were observed between resistant and sensitive samples (*p* < 0.01). The increase in arsenic, iron, and aluminum, along with the decrease in zinc, demonstrated a significant association with vancomycin resistance (*p* < 0.001). Levels of lead, cadmium, mercury, zinc, and iron exceeded permissible limits (*p* < 0.05). The Target Hazard Quotient (THQ) for cadmium indicated a high non-carcinogenic risk, while the Target Risk (TR) for arsenic fell within the carcinogenic range. Accumulation of heavy metals has the potential to impact antibiotic resistance in milk, underscoring the imperative to control arsenic residues for national safety.

## 1. Introduction

In recent years, environmental factors have significantly influenced the induction of diseases at both national and global levels, particularly in developing countries. Cow’s milk is extensively consumed across Iran and worldwide, both in its pure form and as part of dairy products, owing to its high nutritional value. This necessitates a thorough evaluation of its nutritional health [1,2]. It is crucial to note that milk and its derivatives can become contaminated under specific conditions, including water and animal feed contamination with environmental pollutants, sewage effluents, and industrial waste materials, resulting in lead and cadmium contamination [3,4].

These heavy metals accumulate in milk and readily enter the human body. Studies indicate that the concentration of heavy metals in milk depends on their levels in the soil [5]. The severe consequences of heavy metal exposure include neurological symptoms, cancers, nutrient deficiencies, hormonal imbalances, cardiovascular disorders, damage to vital tissues, allergies, asthma, viral infections, weakened immune systems, genetic damage, premature aging, and even death [6,7,8].

*Staphylococcus aureus* is a component of the normal flora of human and warm-blooded animal skin and mucous membranes [9]. It is one of the most common infectious agents in humans, causing diseases such as bacteremia, infective endocarditis, skin infections, lung infections, gastroenteritis, meningitis, and urinary tract infections, depending on the site of infection. Antibiotic therapy is the primary treatment for these infections [10]. Routes of *Staphylococcus aureus* contamination in milk include the transfer of contamination from milking diseased animals with mastitis and secondary contamination with *Staphylococcus aureus* present in the skin (Commensal) [11].

The swift progress in livestock and poultry farming has raised significant concerns about the environmental impact of the widespread use of antibiotics and heavy metals. Antibiotics, crucial for treating diseases like mastitis in dairy cows, are now being misused due to inadequate hygiene practices, irrational prescriptions, and the neglect of essential withdrawal periods. This misuse substantially contributes to the alarming rise of antibiotic resistance, posing a serious threat to both animal and human health.

However, the complex relationship between antibiotic resistance genes (ARGs), microorganisms, and environmental factors remains unclear. In a study, the presence of cadmium (Cd) and sulfadiazine (SD) was investigated, revealing that Proteobacteria and bacterial phyla dominate under these conditions. It is noteworthy that mobile genetic elements (MGEs), especially intI1, profoundly influence ARGs, playing a crucial role in their dissemination and expression [12]. Additionally, a recently discovered pathway illustrates how pathogens produce biofilms to thrive in challenging environments. These biofilms significantly impact antimicrobial resistance (AMR) and heavy metal tolerance. The correlation between biofilm formation and AMR has been observed for most cephalosporins, aminoglycosides, and fluoroquinolones, with biofilm-producing bacteria exhibiting greater tolerance to various metal concentrations. The results of the study suggest that pathogenic genes isolated from dairy and non-dairy foods exhibit high levels of AMR, a clear inclination for biofilm formation, and tolerance to heavy metals, posing an imminent threat to public health [13,14]. In this context, the transfer of pollutants through the food chain is more significant than through water [15].

Therefore, the high density of livestock in the Kurdistan Province initiates the assessment of health and safety standards related to milk production in cattle farms. Furthermore, given the suspended particles and specific mineral conditions in the region’s soil, our attention has been drawn to reports detailing elevated levels of heavy metals. Consequently, our assessment will specifically examine milk from these dairy farms, focusing on the presence of vancomycin-resistant *Staphylococcus aureus* and accumulated levels of heavy metals. Additionally, measuring the concentration of heavy metals in water and cattle feed aims to elucidate the transmission pathways of these pollutants.

## 2. Materials and Methods

In the second quarter of 2023, 160 raw milk samples were gathered from 20 industrial cattle farms located across five geographical areas in the Kurdistan province. These samples were carefully transported to the laboratory in sterile containers, each with a volume of 40 mL. Subsequently, standard microbiological methods were employed to conduct tests aimed at identifying the presence of *Staphylococcus aureus*. The examination encompassed procedures such as Gram staining, catalase testing, coagulase assessment, and evaluation of mannitol sugar fermentation. The steps of the work were approved by the ethics committee in the research of the Islamic Azad University of Sanandaj with the number IR.IAU.SDJ.REC.1402.011.

### 2.1. Milk Sampling

A random cluster sampling method was employed to gather milk samples from cattle farms in the Kurdistan Province. The predominant cattle breed in these industrial farms was Holstein. The milk collection process involved a minimum of five samplings from each dairy farm, conducted in two shifts—morning and afternoon—on consecutive days for each industrial cattle farm.

### 2.2. Isolate and Identify Staphylococcus aureus

To conduct the microbiological analysis, 25 µL of each sample was applied to the blood agar growth medium from the Candalab brand, a product of Spain. The cultures were then incubated at 37 degrees Celsius for 18 h. Following incubation, single colonies exhibiting positive hemolysis, suspected to be *Staphylococcus aureus*, were selected from each sample (typically two to three colonies on average). Subsequently, each chosen colony underwent separate culturing in a blood agar growth medium, utilizing a Quadrant Streak Pattern, and was placed in an incubator at 37 °C for another 18 h. The presence of hemolysis was indicated by a change in the color of the growth medium to yellow. Gram staining procedures were then executed, revealing *Staphylococcus aureus* under a microscope at 100× magnification, exhibiting a characteristic grape cluster arrangement [16].

### 2.3. Biochemical Tests for the Diagnosis of Staphylococcus aureus

For the biochemical tests aimed at diagnosing *Staphylococcus aureus*, the catalase test involved the addition of a drop of 3% hydrogen peroxide to the slide containing the suspicious colony. Positive samples exhibited bubbling within 5–10 s, while the absence of bubbles indicated the absence of catalase [17].

The coagulase slide test commenced with the slow mixing of the suspicious colony with a drop of physiological serum on the slide. Subsequently, a drop of rabbit plasma diluted with EDTA was added and thoroughly mixed. A positive result, signifying aggregation of cocci within 5–10 s, was observed. Following this, the coagulase tube test involved mixing 0.5 mL of diluted rabbit plasma with EDTA at a 1:5 ratio, dissolving several colonies in the resulting milky-colored suspension. Incubating the tubes at 37 °C for 4 h resulted in a positive outcome if clot formation occurred within this time frame, indicating a malignant strain. If no clot formed within 4 h, the tubes were incubated for an additional 24 h and rechecked for clot formation. Failure to form a clot after 24 h suggested a non-pathogenic strain and a negative result [18].

The mannitol salt agar test involved culturing a suspicious colony on an MSA growth medium from the Candalab brand and incubating it for 18 h at 37 °C. Pathogenic *Staphylococcus* strains were identified by their ability to change the color of the MSA growth medium to yellow.

To perform the antibiotic sensitivity test using Kirby–Bauer’s disk diffusion method, a suspension was prepared from positive sample colonies based on the McFarland half-standard. This suspension was cultured on Mueller–Hinton agar (MHA), and a 30-microgram Vancomycin antibiotic disk from Padtanteb company was placed on the MHA with sterile forceps at a specified distance. The culture was then incubated at 35 ± 2 °C for 24 h. The diameter of the zone of bacterial growth inhibition was measured and compared according to the Clinical and Laboratory Standards Institute (CLSI) standard [19,20].

### 2.4. ICP-MS Method Determination

For the determination of metals concentrations in milk, food, and water using the ICP-MS method, each sample was acid-digested by adding 5 mL of it to a sterile falcon tube containing 5 mL of HNO_3_, then allowing it to stand for 24 h. The tube was then heated and strained to ensure dissolution, and the sample volume was increased to 15 mL by adding distilled water. The resulting samples were analyzed using an Agilent 7500 ICP-MS machine; Sciex, USA (2001), with the final results calculated considering the dilution factor (Table 1).

In the final assessment, the safety and health implications for consumers were gauged by determining the Target Hazard Quotient (THQ) and cancer risk (TR) values for heavy metals, utilizing the prescribed formulas [21].


$$\mathrm{THQ}=\frac{\mathrm{EFr}\times \mathrm{ED}\times \mathrm{FIR}\times \mathrm{MC}}{\mathrm{RFD}\times \mathrm{BW}\times \mathrm{TA}}\times 10^{-3}$$


Efr = 365 days/year, ED = 70 years, FIR = 148 g/person/day, MC = mg/kg, RfD = Cd (0.0001 mg/kg); Pb (0.0035 mg/kg); Hg (0.0003 mg/kg); Al (1 mg/kg); Fe (0.7 mg/kg); Zn (0.3 mg/kg); As (0.0003 mg/kg), BW = 70 kg, TA = 25,550 days.
$$\mathrm{TR}=\frac{EFr\times ED\times FIR\times MC\times CSF0}{BW\times TA}\times 10^{-3}$$

EF = 365 day/year, ED = 70 years, FIR = 148 g/person/day, MC = mg/kg, CSF = Cd (6.1 mg/kg/day); Pb (0.0085 mg/kg/day); Hg (1.5 mg/kg/day); Zn (0.3 mg/kg/day); Al (0.02 mg/kg/day); Fe (1.5 mg/kg/day); As (1.5 mg/kg/day), BW = 70 Kg, TA = 25,550 days.

### 2.5. Data Analysis

The data analysis was performed using Prism, and descriptive statistical indicators, including frequency and relative frequency for qualitative variables, as well as mean ± standard deviation for quantitative variables, were employed. The normality of data distribution was assessed using the Shapiro–Wilk test. One-sample t-tests and Wilcoxon tests were utilized to compare the concentrations of heavy metals with the permitted limits. In instances where the assumption of normality was not confirmed, the Mann–Whitney test was employed. The comparison of heavy metal concentrations employed two-sample t-tests. Correlation coefficients, both Pearson and Spearman, were calculated to assess relationships between variables. The significance level for this study was set at *p* < 0.05.

## 3. Results

A total of 160 raw milk samples were analyzed, revealing that 14 (8.75%) tested positive for *Staphylococcus aureus*. Among these positive samples, 10 (71.42%) were determined to be sensitive to vancomycin, while 4 (28.58%) exhibited resistances to vancomycin.

Table 2 provides an overview of the average concentrations of essential and heavy metal in milk samples categorized as either sensitive or resistant to vancomycin. The vancomycin-resistant samples exhibited higher concentrations of arsenic (As) (*p* < 0.001) and sodium (Na) (*p* = 0.04), compared to the vancomycin-sensitive samples. Conversely, the concentrations of iron (Fe) (*p* = 0.03), zinc (Zn) (*p* = 0.005), and sulfur (S) were higher in vancomycin-sensitive milk samples than in vancomycin-resistant samples (Table 2).

The discrepancy in the number of metals and heavy metals between *Staphylococcus aureus* samples resistant and sensitive to vancomycin was found to be statistically significant (Figure 1).

Figure 2 illustrates the correlation coefficients between metal and heavy metal and vancomycin sensitivity. Positive and significant correlations were observed, including As concentration with Ca (r = 0.77, *p* = 0.01), Fe with Al (r = 0.67, *p* = 0.03), Mg with Ca (r = 0.80, *p* = 0.01), Mg with k (r = 0.90, *p* < 0.001), Mg with Na (r = 0.70, *p* = 0.02), Ca with k (r = 0.67, *p* = 0.03), Ca with p (r = 0.71, *p* = 0.02), K with Na (r = 0.92, *p* < 0.001) and P with S (r = 0.93, *p* < 0.001) (Figure 2).

Figure 3 illustrates the correlation between samples infected with vancomycin-resistant *Staphylococcus aureus* and the concentration of metal and heavy metal. Positive and significant correlations were observed, such as the correlation coefficient between As concentration with Cd (r = 0.98, *p* = 0.02), Cd with S (r = 0.96, *p* = 0.04), Pb with Fe (r = 0.97, *p* = 0.03), Fe with k (r = 0.99, *p* < 0.001), Zn with Mg (r = 0.97, *p* = 0.03), Mg with S (r = 0.96, *p* = 0.04) and P with Ca (r = 0.98, *p* = 0.02). On the other hand, negative and significant correlations were found, such as the correlation coefficient between As concentration with P (r = −0.99, *p* < 0.001), Zn with Na (r = −0.97, *p* = 0.03), Mg with Na (r = −0.99, *p* < 0.001), and Na with S (r = −0.97, *p* = 0.03). In the vancomycin-resistant samples, the concentration of Hg was not significantly different from that of the other metals (Figure 3).

Table 3 reveals that the levels of heavy metals in the food and water used in the examined cattle farms indicate that the concentrations of lead, mercury, iron, and aluminum in the animal food surpass those in the water, while those of arsenic do not. Consequently, it appears that the primary pathway for heavy metal intake is through animal feed (Table 3).

Table 4 presents the average concentrations of heavy metals in milk within permissible limits. The average concentration of As in sensitive samples was below the permissible limit; however, in resistant samples, it exceeded the permissible threshold. The average concentration of Pb in sensitive samples surpassed the permissible limit; nevertheless, in resistant samples, it did not exhibit a significant difference from the permissible limit. Both Cd and Hg levels in both resistant and sensitive samples exceeded the permissible limits, with these differences being statistically significant. Additionally, the levels of Fe and Zn were higher than the permissible limit in vancomycin-sensitive samples, while Al exceeded the permissible limit in resistant samples (*p* < 0.05).

As delineated in Table 5, the Target Hazard Quotient (THQ) and cancer risk (TR) metrics are applied to assess heavy metals. The permissible levels of Pb in milk are 0.02, Cd 0.0026, As 0.1, and Hg 0.01, Fe 0.037, Zn 0.328, Al 0.5, as documented by Codex (2007) [22,23].

Applying the aforementioned formula, the calculated THQ values for As, Cd, Pb, Hg, Fe, Zn, and Al are 0.35942, 1.58571, 0.04228, 0.66247, 0.000875, 0.027274, and 0.00133, respectively. The results revealed that only the concentration of Cd in milk exceeded the threshold of one, warranting safety warnings.

Regarding the TR values, results were obtained as follows: As (161.74 × 10^−6^), Cd (967.285 × 10^−6^), Pb (1.26 × 10^−6^), Hg (298.11 × 10^−6^), Fe (919.714 × 10^−6^), Zn (2454.68 × 10^−6^) and Al (26.64 × 10^−6^). The findings indicated that Cd, As, Hg, Fe, Zn and Al fell within the high-risk range, while Pb was categorized in the medium-risk range. TR values below 10^−6^ suggest a low carcinogenic risk, those ranging between 10^−6^ and 10^−4^ indicate a moderate carcinogenic risk, and values between 10^−3^ and 10^−1^ signify a very high carcinogenic risk [21,24].

## 4. Discussion

In a recent comprehensive study, an assessment of essential and heavy metals in milk, water, and animal food was conducted. Subsequently, the results of this evaluation were correlated with the observed vancomycin resistance in *Staphylococcus aureus* isolates. Furthermore, to enhance these findings, the study ensured the safety of the milk produced for consumers by meticulously examining parameters such as TR and THQ. This comprehensive approach not only provides a diverse range of information about the sanitary conditions of the region’s milk, but also presents a synthesis of findings that is unparalleled by any other study. Recent studies have revealed an increase in vancomycin resistance to 28.58%, contrasting with the 0% reported in the 2012 Goldstein study, 9.1% by Corey et al. in 2015, and 23.8% by Al-Saadi in 2017 [25,26,27]. These findings underscore the escalating antibiotic resistance due to the indiscriminate use of antibiotics.

Notably, average concentrations of As, Fe, Zn, Al, and Na significantly differed between vancomycin-resistant and vancomycin-sensitive samples. The rise in concentrations of As and Al, along with a decline in Zn and Fe concentrations, pointed towards an increased resistance to vancomycin in *Staphylococcus* strains isolated from milk. In this context, Tahmasebi et al. (2021) highlighted iron as a noteworthy factor inducing sensitivity to vancomycin, while Corey et al. (2015) demonstrated 100% resistance of *Staphylococcus aureus* isolates to lead, though recent studies failed to establish a clear link between resistance and lead accumulation [27,28].

A substantial correlation between cadmium and arsenic and an increase in antibiotic resistance was observed. However, this correlation was more pronounced for arsenic compared to cadmium, signifying a greater impact of arsenic on antibiotic resistance genes [29]. Despite reports of increased arsenic pollution in Kurdistan, the elevated arsenic levels and their association with vancomycin resistance have been justified [8,30].

The heightened As levels in vancomycin-resistant samples in this study suggest prolonged exposure to Cd and As, contributing to antibiotic resistance in bacteria. The Cd resistance gene in *Staphylococcus aureus* acts independently or in tandem with genes resistant to other metals, such as Pb, Hg, and Zn, as well as antibiotic resistance genes. Long-term exposure to Cd and Pb can exacerbate this resistance [31].

In this study, a noteworthy cause of vancomycin resistance was the decline in Zn concentration. Vancomycin induces the detachment of Zn from bacteria, leading to energy depletion and bacterial demise [32,33]. Consequently, the decrease in Zn concentration is considered a contributing factor to increased resistance to vancomycin. Moreover, vancomycin relies on Zn for its effectiveness, and a reduction in Zn is linked to an escalation in resistance.

Additionally, the reduction in Fe concentration emerged as another factor contributing to vancomycin resistance in this study. Vancomycin prompts the detachment of iron from bacteria, resulting in cell death due to energy depletion. Therefore, the decrease in iron concentration is considered a factor contributing to heightened resistance and insensitivity to vancomycin [34].

In summary, the study revealed a significant correlation between heavy metal concentrations and antibiotic resistance, particularly for arsenic and cadmium. Prolonged exposure to these metals was associated with an increase in antibiotic resistance in bacteria. Additionally, the decrease in zinc and iron concentrations contributed to the development of vancomycin resistance.

In a recent study, the average Pb concentration in positive samples was estimated at 0.7 mg/kg, surpassing the permissible limit of 0.2 mg/kg. Factors contributing to elevated lead levels in milk include the consumption of contaminated soil through grazing, soil mixing with fodder during preparation, and low consumption of mineral supplements. Additionally, casein’s high affinity for lead is considered a significant factor in the increased Pb content in dairy products [35]. Due to its slow excretion rate, lead has the potential for accumulation. Atmospheric suspended matter deposits, vehicle exhaust, and urban runoff are other sources of lead pollution in milk and dairy products.

The average Cd concentration in positive samples was 0.08 mg/kg, exceeding the permissible limit in milk. Contact with soil and chemical fertilizers is among the major sources of Cd pollution in milk and dairy products. Studies have reported a positive correlation between Cd and Pb in milk, as well as their concentrations in soil and animal food, but their correlation with consumed water has been observed [36].

The average As concentration in positive samples was 0.05 mg/kg, lower than the permissible limit of 0.1 mg/kg in milk. The permissible limit for mercury in milk was 0.01 mg/kg, indicating an exceedance. Therefore, in addition to various factors such as geographical location, agricultural practices, and environmental conditions, physiological effects on animals, including increased age in cows, should be considered, leading to the accumulation of essential or toxic substances in milk [37,38].

Moreover, the results indicate that heavy metal levels in food and water reflect the increased transfer of heavy metals from food to cattle milk due to the interaction of forage. Heavy metals such as lead, mercury, iron, and aluminum, but not arsenic, have been found to enter the milk of dairy farms in this region through the consumption of contaminated food.

The Target Hazard Quotient (THQ) for Cd, As, and Pb was found to be safe and non-hazardous in the current study. However, the THQ for Cd exceeded one, indicating a high non-carcinogenic risk for this metal in the collected milk samples. THQ has an inverse relationship with age, and therefore, children, due to higher milk consumption and lower body weight, are at the highest risk of non-carcinogenic risk [39].

In the current study, As was evaluated as having a high carcinogenic risk. Intoxication with toxic heavy metals depends on their daily consumption. Due to the higher consumption of milk compared to other foods in sensitive age groups such as children and the elderly, poisoning with toxic heavy metals is a serious concern [39].

## 5. Conclusions

The findings of this study validate the correlation between vancomycin antibiotic resistance and heavy metals, including As, Al, Zn, and Fe. This correlation suggests that an increase in the concentration of toxic heavy metals may lead to heightened resistance. Furthermore, the Total Hazard Quotient (THQ) and Target Risk (TR) values for Cd and As in this study indicate an impact on the food safety of milk. Additionally, the association between the concentration of heavy metals in milk, excluding As, was determined based on the food consumed by livestock.

Given that milk is considered a wholesome dietary element that influences growth, increased scrutiny of water sources, livestock feed, their positioning, and the use of environmental antibiotics is deemed a priority. Furthermore, the prudent management of antibiotic usage is emphasized, considering the presence of a social health cycle, to regulate the unchecked growth of resistance to vancomycin—an antibiotic of utmost importance.

## Figures and Tables

**Figure 1 animals-14-00148-f001:**
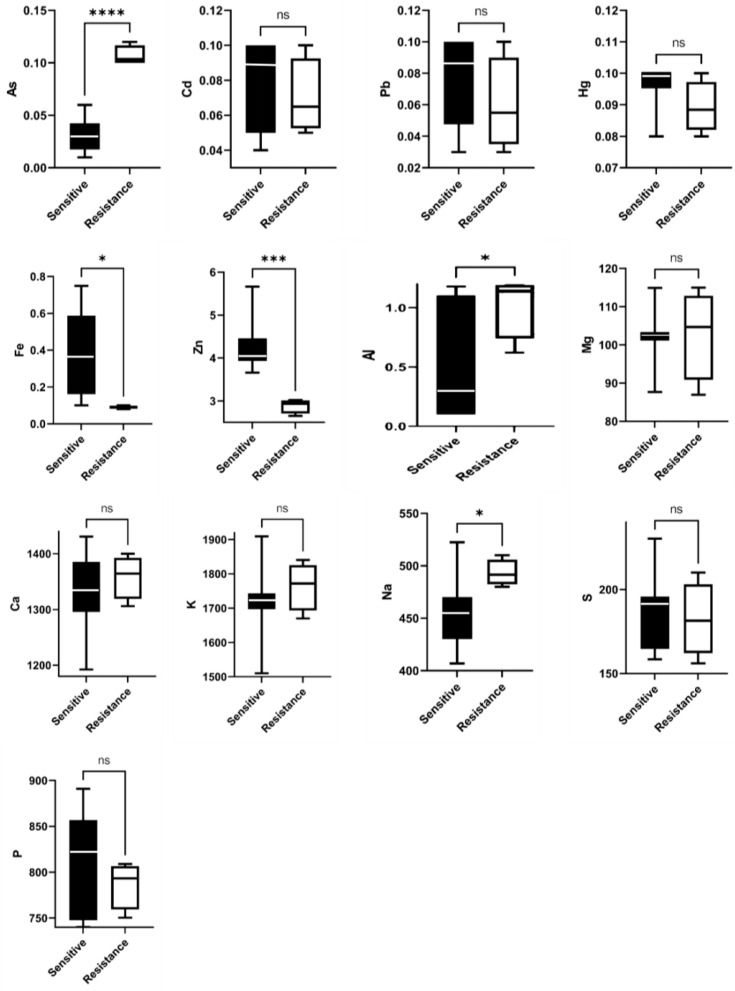
Box plot average concentration of metal and heavy metal: ns: the difference between the two groups is not significant. * The difference between the two groups is significant at the 0.05 level. *** The difference between the two groups is significant at the 0.001 level. **** The difference between the two groups is significant at the 0.0001 level.

**Figure 2 animals-14-00148-f002:**
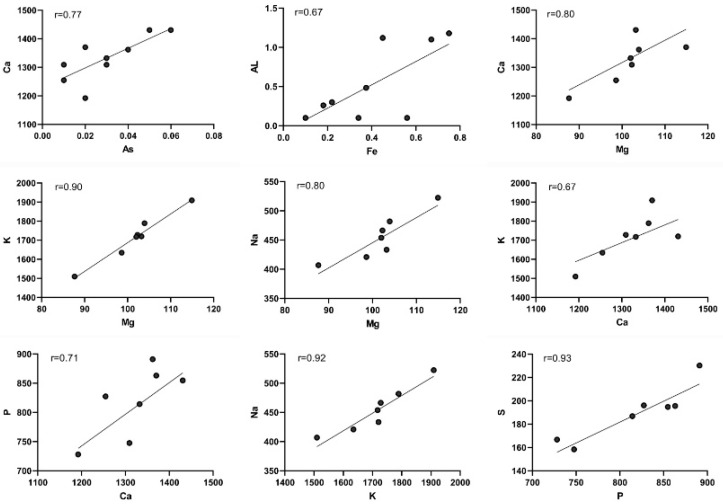
Correlation between metals in vancomycin-sensitive samples.

**Figure 3 animals-14-00148-f003:**
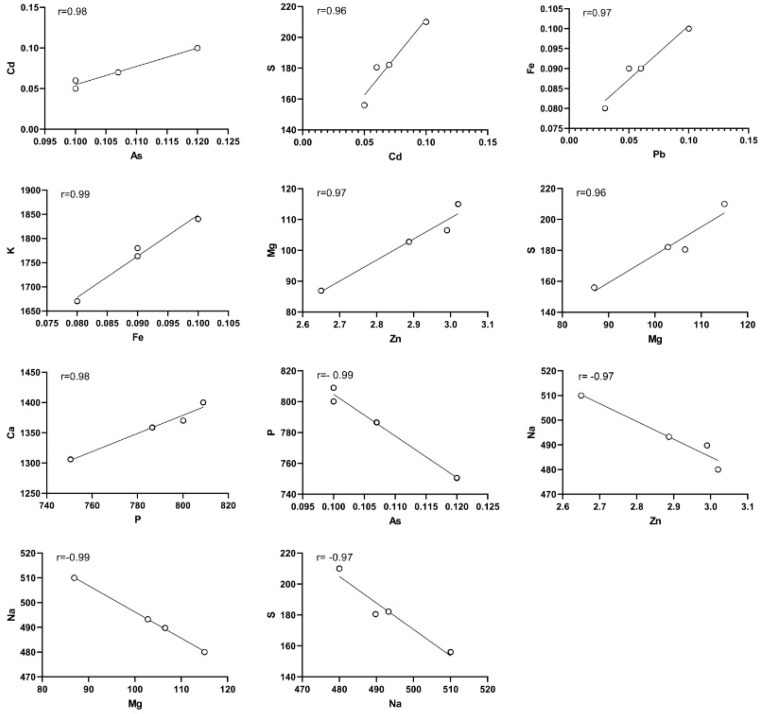
Correlation between metals in vancomycin-resistant samples.

**Table 1 animals-14-00148-t001:** Specifications of the ICP-MS device.

Parameters	Value/Type
RF generator Power	1200 W
RF frequency	Resonance frequency: 24 MHz
Plasma, auxiliary, and nebulizer gas	Argon
Plasma gas flow rate	2/12 (L/min)
Auxiliary gas flow rate	0/8 (L/min)
Nebulizer gas flow rate	0/8 (L/min)
Sample uptake time	260 total (S)
Measurement replicate	3
Type of detector solid state	CCD
Type of spray chamber cyclonic	Modified Lichte

**Table 2 animals-14-00148-t002:** Comparison of the average concentration of metal and heavy metal in vancomycin-resistant and -sensitive positive samples.

Metals	Group	Mean (SD)	Mean Difference (CI 95%)	Median (Min-Max)	Statistical Test	*p*-Value
As	Sensitive	0.03 (0.02)	(0.02, 0.04)	0.03 (0.01, 0.06)	t = 8.70	<0.001
	Resistance	0.11 (0.01)	(0.09, 0.12)	0.10 (0.10, 0.12)		
Cd	Sensitive	0.08 (0.03)	(0.06, 0.10)	0.09 (0.04, 0.10)	z = 0.44	0.66
	Resistance	0.07 (0/02)	(0.04, 0.10)	0.07 (0.05, 0.10)		
Pb	Sensitive	0.07 (0.03)	(0.05, 0.10)	0.09 (0.03, 0.10)	z = 0.74	0.46
	Resistance	0.06 (0.03)	(0.01, 0.11)	0.06 (0.03, 0.10)		
Hg	Sensitive	0.10 (0.01)	(0.09, 0.10)	0.10 (0.08, 0.10)	z = 1.72	0.08
	Resistance	0.09 (0.01)	(0.08, 0.10)	0.09 (0.08, 0.10)		
Fe	Sensitive	0.37 (0.23)	(0.21, 0.54)	0.36 (0.10, 0.75)	t = 2.40	0.03
	Resistance	0.09 (0.01)	(0.08, 0.10)	0.09 (0.08, 0.10)		
Zn	Sensitive	4.26 (0.61)	(3.82, 4.70)	4.05 (3.66, 5.66)	z = 2.84	0.005
	Resistance	2.89 (0.17)	(2.62, 3.15)	2.94 (2.65, 3.02)		
Al	Sensitive	0.48 (0.46)	(0.15, 0.82)	0.28 (0.10, 1.18)	Z = 2.01	0.04
	Resistance	1.02 (0.27)	(0.59, 1.46)	1.14 (0.62, 1.2)		
Mg	Sensitive	102.01 (6.59)	(97.29, 106.72)	102.27 (87.66, 114.93)	z = 0.71	0.48
	Resistance	102.80 (11.77)	(84.08, 121.52)	104.65 (86.90, 115.00)		
Ca	Sensitive	1332.39 (73.36)	(1279.91, 1384.87)	1332.39 (1192.20, 1430.73)	t = 0.67	0.52
	Resistance	1358.67 (39.21)	(1296.28, 1421.06)	1363.34 (1306.00, 1400.01)		
K	Sensitive	1717.57 (101.14)	(1645.22, 1789.92)	1720.38 (1509.99, 1909.35)	t = 0.82	0.43
	Resistance	1763.43 (70.50)	(1651.25, 1875.61)	1771.72 (1670.00, 1840.29)		
Na	Sensitive	453.98 (33.24)	(430.2, 477.75)	453.98 (406.95, 522.48)	t = 2.25	0.04
	Resistance	493.26 (12.49)	(473.38, 513.14)	491.52 (480.00, 510.00)		
S	Sensitive	186.90 (21.61)	(171.45, 202.36)	190.86 (158.37, 230.28)	t = 0.37	0.72
	Resistance	182.20 (22.07)	(147.07, 217.32)	181.40 (156.00, 210.00)		
P	Sensitive	814.32 (55.82)	(774.39, 854.25)	820.87 (728.16, 891.03)	t = 0.94	0.37
	Resistance	786.55 (25.75)	(745.58, 827.52)	793.35 (750.50, 809.01)		

The gray color indicate a statistically significant difference between vancomycin resistance and sensitivity (*p* < 0.05).

**Table 3 animals-14-00148-t003:** Content of heavy metals in the food and water used in cattle farms in Kurdistan province.

Metals	Parameters	Mean (SD)	Max-Min	Statistical Test	*p*-Value
As	Food	0.05 (0.01)	0.01–0.06	F = 5.85	<0.001
	Water	0.82 (0.04)	0.60–1.0		
Cd	Food	0.09 (0.04)	0.12–0.10	F = 0.63	0.05
	Water	0.08 (0.02)	0.09–0.01		
Pb	Food	0.09 (0.04)	0.12–0.05	F = 5.34	<0.001
	Water	0.04 (0.01)	0.02-0.09		
Hg	Food	0.3 (0.02)	0.32–0.05	F = 4.37	0.01
	Water	0.06 (0.01)	0.08–0.02		
Fe	Food	0.46 (0.24)	0.86–0.28	F = 5.38	0.04
	Water	0.07 (0.02)	0.15-0.06		
Zn	Food	3.12 (0.36)	3.78–3.52	F = 3.85	0.48
	Water	2.15 (0.22)	3.01–0.68		
Al	Food	1.27 (0.51)	1.63–0.68	F = 4.25	0.032
	Water	1.02 (0.27)	0.76–0.37		

The gray color indicates a statistically significant difference in the concentration of heavy metals between water and food (*p* < 0.05).

**Table 4 animals-14-00148-t004:** Comparison of the average concentration of heavy metals in samples resistant and sensitive to vancomycin with the permitted level.

Metals	Group	Mean	Mean Difference (CI 95%)	Limit	Statistical Test	*p*-Value
As	Sensitive	0.03	(0.02, 0.04)	0.10	t = 13.56	0.012
	Resistance	0.11	(0.09, 0.12)	0.10	t = 14.43	0.025
Cd	Sensitive	0.08	(0.06, 0.10)	0.0026	w = 0.084	0.002
	Resistance	0.07	(0.04, 0.10)	0.0026	t = 6.24	0.008
Pb	Sensitive	0.07	(0.05, 0.10)	0.02	w = 0.067	0.002
	Resistance	0.06	(0.01, 0.11)	0.02	t = 2.72	0.07
Hg	Sensitive	0.10	(0.09, 0.10)	0.01	w = 0.08	0.002
	Resistance	0.09	(0.08, 0.10)	0.01	t = 19.28	<0.001
Fe	Sensitive	0.37	(0.21, 0.54)	0.037	t = 4.61	0.001
	Resistance	0.09	(0.08, 0.10)	0.037	t = 12.98	0.001
Zn	Sensitive	4.26	(3.82, 4.70)	0.328	w = 0.08	0.002
	Resistance	2.89	(2.62, 3.15)	0.328	t = 30.50	<0.001
Al	Sensitive	0.48	(0.15, 0.82)	0.50	t = 20.34	0.001
	Resistance	1.02	(0.59, 1.46)	0.50	t = 15.29	0.002

**Table 5 animals-14-00148-t005:** THQ and TR toxic heavy metals in studied raw milk.

Parameters	As	Cd	Pb	Hg	Fe	Zn	Al
THQ	0.35942	1.58571	0.04228	0.66247	0.000875	0.027274	0.00133
TR	161.74 × 10^−6^	967.285 × 10^−6^	1.26 × 10^−6^	298.11 × 10^−6^	919.714 × 10^−6^	2454.68 × 10^−6^	26.64 × 10^−6^

## Data Availability

The datasets used and/or analyzed during the current study are available from the corresponding author upon reasonable request.

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
