# Peer review of "The Impact of Metal and Heavy Metal Concentrations on Vancomycin Resistance in *Staphylococcus aureus* within Milk Produced by Cattle Farms and the Health Risk Assessment in Kurdistan Province, Iran"

_animals, 2024, doi:10.3390/ani14010148_

Round 1
Reviewer 1 Report (Previous Reviewer 1)
Comments and Suggestions for Authors
Dear authors,
The present document solid and clear information for one of the main problems related with quality milk and their risks for feeding people.
Please, check the comments registered into the document.

Author Response
I am infinitely grateful since the honorable referee has devoted time to reviewing the article. The requested revisions have been incorporated into the text.
Reviewer 2 Report (Previous Reviewer 2)
Comments and Suggestions for Authors
It is better to move the drawing to section 2
Author Response
I am immensely grateful to the esteemed referee for dedicating time to review the article. The graphical abstract has been relocated from the main text to the appendices, to be considered in the article's list. Additionally, references have been double-checked against the text.
Reviewer 3 Report (Previous Reviewer 3)
Comments and Suggestions for Authors
Paper is acceptable
Author Response
I appreciate the esteemed referee for the time spent reviewing the article.
This manuscript is a resubmission of an earlier submission. The following is a list of the peer review reports and author responses from that submission.
Round 1
Reviewer 1 Report
Comments and Suggestions for Authors
Dear authors,
The topic of the present paper is one of the main problems for dairy farms due to reduce the quality of the milk and increase the risk of public health by mean presence of VRSA or simply MDR bacteria.
It is necesary to attended the comments described in the document and answer the questions

Author Response
Reviewer#1
Dear authors,
The topic of the present paper is one of the main problems for dairy farms due to reduce the quality of the milk and increase the risk of public health by mean presence of VRSA or simply MDR bacteria.
Response to the comments by the Reviewer
I appreciate the positive viewpoint of the learned reviewer and thank you for your feedback on the article revisions. Your insightful comments have been individually integrated into the revised text.
It is necesary to attended the comments described in the document and answer the questions
2.1. Thank you very much for your attention. Change has been made.
Italic writing of the scientific name in the entire text
Thanks. Changes have been made.
HNO3
Thanks. Please refer to line 160.
Figure 3 should be black and white.
Thank you for the recommendation from the knowledgeable arbitrator. All figures have been converted to black and white.
Reviewer 2 Report
Comments and Suggestions for Authors
Dear authors!
Thank you for providing an interesting manuscript for publication in the journal "Animals".
However, significant inaccuracies and errors have been discovered that require attention and elimination:
1. Сheck the location and correctness of all the drawings (should they be there)?
2. In the name of "heavy metal residues ", where do you specify?
3. line 68-71 : The statement resistance to heavy metals is incorrect. There is no such stability.
4. In what form are antibiotics used on farms? In the form of medications (intradermally) or feed additives?
5. What is the reason for the choice of such studies?
6. And is it allowed in Iran to use antibiotics in the production of milk and dairy products (add and in accordance with the legislation)?
7. It is necessary to highlight the ethics protocol separately.
8. And a separate chapter on milk sampling.
9. It is not clear how the milk was taken. From each head? from the milk tank? What is the breed of cows? How often (1 time from the farm or more often)?
10. Was the combined milk used per day or a specific act (for example, morning or lunch)?
11. Were samples taken everywhere at the same time or at different times?
How is the administration of antibiotics or heavy metals in milk related?
12. You don't think so. what is it primarily related to animal feeding: feed and water?
13. Nowhere in the manuscript are the ration compositions of all farms and the content of heavy metals in them. Or the content in the water, which may be their source.
14. The article had to be framed in accordance with the requirements of the journal.
I think that the article is not enough for publication in the Q1 journal.
Author Response
Reviewer #2
Dear authors!
Thank you for providing an interesting manuscript for publication in the journal "Animals".
However, significant inaccuracies and errors have been discovered that require attention and elimination:
I'm grateful for the positive feedback from the reviewer. I've carefully addressed each of your valuable comments throughout the article.
- Сheck the location and correctness of all the drawings (should they be there)?
Thank you for the careful opinion of the learned Reviewer. Changes have been made
- In the name of "heavy metal residues ", where do you specify?
Thanks for your attractive comment. heavy metal accumulation in raw animal products. Since the term accumulation is used for metals, the title and text were modified.
- line 68-71 : The statement resistance to heavy metals is incorrect. There is no such stability.
Thank you for your comment, dear referee. The sentence was corrected. Please refer to the lines 75-89.
- In what form are antibiotics used on farms? In the form of medications (intradermally) or feed additives?
Antibiotics are mostly used in the treatment of diseases, including mastitis in dairy cows. Poor hygiene conditions, irrational prescription of antibiotics, and failure to observe withdrawal periods can contribute to the development of effective antibiotic resistance.
- What is the reason for the choice of such studies?
I appreciate the valuable question from the learned reviewer. Considering that every research endeavor aims to illuminate the current state of affairs and assess forthcoming challenges, this study has been conducted with a focus on the specific issues and conditions prevalent in the Kurdistan region. The objective is to shed light on the health status of the region's milk and to contemplate subsequent monitoring and managerial plans. This is explicitly mentioned in the text.
"The livestock population density in Kurdistan Province has prompted an assessment of the health and safety of the milk produced in these dairy farms. Additionally, considering the suspended particles and mineral conditions in this region, reports regarding the levels of heavy metals have led us to focus on evaluating the health and safety of the milk from these dairy farms. This assessment will specifically concentrate on the presence of vancomycin-resistant Staphylococcus aureus bacteria and the accumulation levels of heavy metals".
- And is it allowed in Iran to use antibiotics in the production of milk and dairy products (add and in accordance with the legislation)?
Thanks for your question. The use of antibiotics in treating animal diseases in Iran, like in other countries, is common. However, antibiotic consumption is associated with an increase in drug resistance and genomic changes due to mutations or adaptation. Examining this resistance is crucial for assessing the current and future health conditions in the livestock industry and human health.
- It is necessary to highlight the ethics protocol separately.
I appreciate the comment from the respected reviewer. A separate paragraph stating the ethical statement has been added to the text of the article.
- And a separate chapter on milk sampling.
Thanks. A section titled "Milk Sampling" has been added.
- It is not clear how the milk was taken. From each head? from the milk tank? What is the breed of cows? How often (1 time from the farm or more often)? Please refer to the Milk Sampling section
- Was the combined milk used per day or a specific act (for example, morning or lunch)?
Please refer to the Milk Sampling section
- Were samples taken everywhere at the same time or at different times?
I express my gratitude for the insightful and valuable comments provided by the esteemed reviewer. In the section titled "Milk Sampling," the article has addressed questions 8, 9, 10, and 11. The sampling process involved collecting milk from Holstein cattle in the industrial dairy farms of Kurdistan Province during both morning and afternoon milking sessions. This procedure was iterated five times for each dairy farm on consecutive days to capture a range of biological and environmental variations in the samples.
How is the administration of antibiotics or heavy metals in milk related?
Various factors and reasons can play a role in the relationship between heavy metal concentration and effective antibiotic resistance. This includes mobile genetic elements (MGEs), especially intI1, and the production of biofilms by bacteria under conditions where resistance to antibiotics increases in conjunction with the rise in heavy metal concentrations. which has been mentioned in various studies
13.EJAZ, Hasan, et al. Multiple antimicrobial resistance and heavy metal tolerance of biofilm-producing bacteria isolated from dairy and non-dairy food products. Foods, 2022, 11.18: 2728.
- Dweba C.C., Zishiri O.T., El Zowalaty M.E. Isolation and Molecular Identification of Virulence, Antimicrobial and Heavy Metal Resistance Genes in Livestock-Associated Methicillin-Resistant Staphylococcus aureus. Pathogens. 2019;8:79. doi: 10.3390/pathogens8020079. [PMC free article] [PubMed] [CrossRef] [Google Scholar]
R1.Qin G., Niu Z., Yu J., Li Z., Ma J., Xiang P. Soil heavy metal pollution and food safety in China: Effects, sources and removing technology. Chemosphere. 2021;267:129205. doi: 10.1016/j.chemosphere.2020.129205. [PubMed] [CrossRef] [Google Scholar]
R2. Junaid K., Ejaz H., Asim I., Younas S., Yasmeen H., Abdalla A.E., Abosalif K.O.A., Alameen A.A.M., Ahmad N., Bukhari S.N.A., et al. Heavy Metal Tolerance Trend in Extended-Spectrum β-Lactamase Encoding Strains Recovered from Food Samples. Int J. Environ. Res. Public Health. 2021;18:4718. doi: 10.3390/ijerph18094718. [PMC free article] [PubMed] [CrossRef] [Google Scholar]
- You don't think so. what is it primarily related to animal feeding: feed and water?
Thank you for your question. As soil plays a fundamental role in ensuring the safety of food substances, and pollutants such as heavy metals directly affect the quality of forage and livestock feed, the transfer of resistance genes from soil to microorganisms becomes a significant factor. Consequently, the relative importance of feed, compared to water, in the health of both livestock and humans can be recognized. In confirmation, the results presented in Table 3 indicate that, except for arsenic, the concentrations of other metals in the feed are higher than in the water consumed by animals, suggesting that these metals are transferred through the animal feed.
15.QIN, Guowei, et al. Soil heavy metal pollution and food safety in China: Effects, sources and removing technology. Chemosphere, 2021, 267: 129205.
- Nowhere in the manuscript are the ration compositions of all farms and the content of heavy metals in them. Or the content in the water, which may be their source.
I appreciate the insightful comment from the reviewer. Based on Table 3, information about heavy metals in the feed and water consumed by animals has been presented. The results indicate the significance of animal feed compared to water in the content of heavy metals.
- The article had to be framed in accordance with the requirements of the journal.
The article has been prepared in accordance with the journal's writing guidelines.
I think that the article is not enough for publication in the Q1 journal.
The article has undergone significant improvement based on the valuable feedback provided by the esteemed reviewer on the initial draft. The findings regarding heavy metal concentrations in the feed and water consumed by cattle have been incorporated. Furthermore, a comprehensive consideration of all heavy metals concerning safety and health parameters has been undertaken, and the relevant information has been added and finalized in Tables 4 and 5.
Reviewer 3 Report
Comments and Suggestions for Authors
This is an interesting study that attempts to link heavy metal milk concentration with vancomycin resistance. as the authors indicated, this is nothing new here as this has been reported and well recognized observation. However, the authors should address the following concerns:
1) They need to highlight the novelty of their research and what separates it from other studies. This needs to be highlighted.
2) Not all of the substances mentioned in the paper are heavy metals; eg., Na, K, Ca, are not heavy metals.
3) Authors need to state the units for their concentrations and be consistent about it; ppm or mg/kg.
4) On page 4 in results, authors need to indicate if "higher" means statistically significant and provide p value.
5) Table 3 states "permitted level". need to cite if it is Codex etc.
6) Line 256: "upsurge" compare to what?
7)
Comments on the Quality of English Language
Authors did a good job here but careful in the use of words like "upsurge"
Author Response
Reviewer #3
This is an interesting study that attempts to link heavy metal milk concentration with vancomycin resistance. as the authors indicated, this is nothing new here as this has been reported and well recognized observation. However, the authors should address the following concerns:
I am very grateful for the positive feedback and insightful comments from the learned reviewer. Your considerations and comments have been precisely incorporated into the text of the article.
1) They need to highlight the novelty of their research and what separates it from other studies. This needs to be highlighted.
I appreciate the insightful comment from the esteemed reviewer. The first paragraph discusses the strengths of the study compared to previous research.
2) Not all of the substances mentioned in the paper are heavy metals; eg., Na, K, Ca, are not heavy metals.
I appreciate the attention of the learned reviewer. The title has been change.
3) Authors need to state the units for their concentrations and be consistent about it; ppm or mg/kg.
Thanks. The unit is considered in mg/kg.
4) On page 4 in results, authors need to indicate if "higher" means statistically significant and provide p value.
Thank you for your comment. The results have been revised on page 4, and significant findings are indicated with p-values.
5) Table 3 states "permitted level". need to cite if it is Codex etc.
Thank you for your comment. Codex cited in the paper (References 22, 23)
- Masson-Matthee, M.D., The codex alimentarius commission and its standards. 2007: Springer.
- FAO/WHO- Food and Agriculture Organization/World Health Organization. Joint FAO/WHO food standards program: Codex committee on contaminants in foods (Editorial amendments to the general standard for contaminants and toxins in food and feed), sixth session, Maastricht, Netherlands, 26-30 march, 2012; CX/CF 12/6/11
6) Line 256: "upsurge" compare to what?
I appreciate the insightful comments from the esteemed reviewer. The sentence structure has been revised.
7)
Comments on the Quality of English Language
Authors did a good job here but careful in the use of words like "upsurge"
Thank you for the reviewer's comment. The language of the paper has been reviewed again.